

# Intervention effect of exercise on working memory in patients with depression: a systematic review

Cong Liu[1], Rao Chen[2], So Mang Yun[1] and Xing Wang[1]

[1] School of Physical Education, Shanghai University of Sport, Shanghai, China
[2] Shanghai I&C Foreign Languages School, Shanghai, China

## ABSTRACT

**Background:** This article aims to systematically evaluate the intervention effect of exercise on working memory in patients with depression.

**Methods:** Six Chinese and English databases were searched for randomized controlled trials (RCTs) about exercise on working memory in patients with depression. PEDro scale was adopted to evaluate the methodological quality of the included articles, GRADEpro scale was employed to evaluate the level of evidence for outcomes, and the Metafor Package in R 4.4.1 was used to analyze the combined effect size, subgroup analyses and publication bias.

**Results:** A total of 15 studies were included. The meta-analysis indicated that exercise had a statistically significant effect on working memory in patients with depression, with an effect size of 0.16 (95% CI [0.03–0.28], $p = 0.02$). Exercise type ($F_{(3,34)} = 1.99$, $p = 0.13$), intervention content ($F_{(1,36)} = 1.60$, $p = 0.22$), and exercise duration ($F_{(1,36)} = 0.05$, $p = 0.83$) did not moderate the effect, whereas exercise intensity showed a moderating effect ($F_{(2,35)} = 8.83$, $p < 0.01$). There was evidence of publication bias in the study results ($t = 2.52$, $p = 0.02$).

**Conclusion:** Exercise can improve the working memory of patients with depression, and its moderating effect is the best when having low-intensity and moderate-intensity. Research plan was registered in international system evaluation platform PROSPERO (https://www.crd.york.ac.uk/PROSPERO/) (CRD42023475325).

Corresponding author
Xing Wang,
wangxing1933@sus.edu.cn

## INTRODUCTION

Depression is a syndrome featuring emotional, cognitive, and somatic symptoms, with high prevalence, recurrence, and suicide rates (*Miret et al., 2013*; *Wen et al., 2022*). Globally, approximately 260 million people suffer from depression (*Liu et al., 2020*), and unipolar depression is projected to become the second leading cause of global disease burden by 2030 (*Mathers & Loncar, 2006*). In China, the prevalence of major depressive disorder is as high as 3.4%, and the lifetime disability rate is as high as 47%, which brings heavy pressure to the society (*Huang et al., 2019*). The cognitive theory of depression suggests that patients with depression have executive function deficits, and show difficulty in removing negative information, maintaining positive information in working memory, and appear information processing bias and memory bias consistent with mood

(*Hong et al., 2022*; *Jenkins, 2019*). Impairment of cognitive function, such as working memory, not only affects the effectiveness of drug treatment for patients with depression, but also increases the risk of recurrence of depression (*Chinese Society of Psychiatry CAoDD, 2020*).

As one of the safe non-medical interventions with few side effects, exercise not only poses a good effect on antidepressant, but also plays a potential role in improving cognitive function. Studies have found that regular exercise can promote neurogenesis, enhance synaptic plasticity, and promote capillary proliferation (*Kim et al., 2019*). Some researchers conducted aerobic exercise of different intensities on patients with depression, and used N-back to evaluate the working memory of patients, whose results found that aerobic exercise of different intensities had selective improvement in the working memory of patients (*Kubesch et al., 2003*). Some studies have also conducted aerobic exercise combined with mental relaxation training in patients with depression and schizophrenia, and the working memory of them was assessed by digit span, whose results showed that the patients' working memory improved significantly (*Oertel-Knöchel et al., 2014*). On the contrary, some studies have also conducted aerobic exercise interventions in patients with depression and evaluated patients' working memory by digital-forward and digital-backward reading, and after exercise, their working memory did not improve (*Krogh et al., 2012*). The reason for the different results may be attributed to the different control elements of exercise intervention such as exercise form, exercise duration, exercise cycle and exercise intensity among different experiments.

By musing on previous studies, it was found that the meta-analyses results of exercise improving working memory in patients with depression were also inconsistent. The meta-analyses of *Sun et al. (2018)* and *Brondino et al. (2017)* found no improvement in working memory in patients with depression after doing exercise. In contrast, the meta-analyses of *Ren et al. (2023)* and *Contreras-Osorio et al. (2022)* indicated that exercise can improve working memory in these patients. However, none of these studies explored the effects of exercise type, exercise intensity, exercise cycle, and intervention content on working memory, despite examining the impact of exercise on it. Therefore, this study aims to further investigate the intervention effects of exercise on working memory in patients with depression. Building upon previous researches, it seeks to clarify the dose-response relationships of various components of exercise on the intervention effects of working memory in an effort to identify the optimal exercise regimen, provide evidence-based recommendations for clinical practice and serve as a theoretical reference for researchers in the field.

## MATERIALS AND METHODS

This study was conducted in accordance with the methods and requirements of the PRISMA statement and the Cochrane Handbook of Work (*Higgins & Green, 2008*; *Page et al., 2021*). Research plan was registered in international system evaluation platform PROSPERO (https://www.crd.york.ac.uk/PROSPERO/) (CRD42023475325). The PICOS framework of this study is shown in Table 1.

**Table 1 PICOS architecture of the effects of exercise intervention on working memory in depressed patients.**

| PICOS | Content |
|---|---|
| Subjects | Patients with depression: meet either the diagnostic criteria of the *International Classification of Disease* (ICD) or *Diagnostic and Statistical Manual of Mental Disorders* (DSM) |
| Intervention | Exercise or exercise based on the intervention of the control group |
| Comparison | Routine treatment, daily life and stretching exercise, *etc.* |
| Outcome indicators | This study included outcome measures related to working memory |
| Study design | RCTs |

## Literature inclusion criteria

The subjects were patients with depression, meeting either the criteria of *International Classification of Disease* (ICD) and the *Diagnostic and Statistical Manual of Mental Disorders Disorders* (DSM). The intervention group consisted of exercise alone or exercise combined with other interventions such as antidepressant medication and cognitive therapy. We included studies that focused on long-term exercise, defined as any "bodily movement produced by skeletal muscles with the expenditure of energy" (*Hillman, Erickson & Kramer, 2008*), sustained for a minimum of 3 weeks. The control group interventions included routine care, daily activities, antidepressant medication, cognitive therapy, and relaxation exercises. Outcome indicators are related to working memory.

## Literature exclusion criteria

The study population was non-depressed patients. The intervention was acute exercise. The outcome indicators in the study did not meet criteria or could not be extracted; the study belonged to literature review, *etc.*; the languages studied were not Chinese or English.

## Literature retrieval strategy

CNKI, Wanfang, PubMed, Embase, The Cochrane Library, and Web of Science databases were searched by two researchers (CL and RC) independently for RCTs on the exercise effect on working memory in patients with depression. Retrieval data started from the establishment date of each database to October 23, 2023. The retrieval method was subject words combined with free words, Boolean operation symbols "AND" and "OR" were used to combine and connect, and were confirmed after repeated pre-checking. If two researchers encountered disagreements, a third researcher (XW) would join in the discussion and make a joint decision. We only collected Chinese articles that were included in the *Core Journal of China*, while for English articles we did not have such limitations. A subsequent supplement was conducted to trace relevant systematic reviews and references of included article for those not having been retrieved, and the specific retrieval strategy is shown in Table 2.

**Table 2** Search strategies for each database.

| Database | Retrieval strategy |
|---|---|
| Cochrane and PubMed | #1 "Exercise" [Mesh] OR "Aerobic exercise" [Title/Abstract] OR "Resistance exercise" [Title/Abstract] OR "High-intensity interval" [Title/Abstract] OR "Yoga" [Title/Abstract] OR "Dance" [Title/Abstract] OR "Taichi" [Title/Abstract] OR "Baduanjin" [Title/Abstract] OR "Wuqinxi" [Title/Abstract] OR "Yijinjing" [Title/Abstract] OR "Walking" [Title/Abstract] OR "Physical and mental exercise" [Title/Abstract] |
| | #2 "Depression"[Mesh] OR "Depressive disorder" [Title/Abstract] OR "Depressive symptom" [Title/Abstract] OR "Emotional depression" [Title/Abstract] OR "Depressive neurosis" [Title/Abstract] OR "Endogenous depression" [Title/Abstract] OR "Deurotic depression" [Title/Abstract] OR "Unipolar depression" [Title/Abstract] |
| | #3 "Memory, Short-Term" [Mesh] OR "Cognition" [Title/Abstract] OR "Cognitive performance" [Title/Abstract] OR "Cognitive" [Title/Abstract] OR "Working memory" [Title/Abstract] OR "Shifting" [Title/Abstract] |
| | #4 Randomized controlled trial [Publication Type] OR "Randomized" [Title/Abstract] OR "controlled" [Title/Abstract] OR "Trial" [Title/Abstract] |
| | #5 #1 AND #2 AND #3 AND #4 |
| Embase | #1 "Exercise" [exp] OR "Aerobic exercise" [ab,ti] OR "Resistance exercise" [ab,ti] OR "High-intensity interval" [ab,ti] OR "Yoga" [ab,ti] OR "Dance" [ab,ti] OR "Taichi" [ab,ti] OR "Baduanjin" [ab,ti] OR "Wuqinxi" [ab,ti] OR "Yijinjing" [ab,ti] OR "Walking" [ab,ti] OR "Physical and mental exercise" [ab,ti] |
| | #2 "Depression"[exp] OR "Depressive disorder" [ab,ti] OR "Depressive symptom" [ab,ti] OR "Emotional depression" [ab,ti] OR "Depressive neurosis" [ab,ti] OR "Endogenous depression" [ab,ti] OR "Deurotic depression" [ab,ti] OR "Unipolar depression" [ab,ti] |
| | #3 "Working memory" [exp] OR "Cognition" [ab,ti] OR "Cognitive performance" [ab,ti] OR "Cognitive" [ab,ti] OR "Working memory" [ab,ti] OR "Shifting" [ab,ti] OR "Cognitive" [ab,ti] |
| | #3 "Randomized controlled trial" [exp] OR "Randomized" [ab,ti] OR "Controlled" [ab,ti] OR "Trial" [ab,ti] |
| | #5 #1 AND #2 AND #3 AND #4 |
| Web of Science | #1 TS = ("Exercise" OR "Aerobic exercise" OR "Resistance exercise" OR "High-intensity interval" OR "Yoga" OR "Dance" OR "Taichi" OR "Baduanjin" OR "Wuqinxi" OR "Yijinjing" OR "Walking" OR "Physical and mental exercise") |
| | #2 TS = ("Depression" OR "Depressive disorder" OR "Depressive symptom" OR "Emotional depression" OR "Depressive neurosis" OR "Endogenous depression" OR "Deurotic depression" OR "Unipolar depression") |
| | #3 TS = ("Cognition" OR "Cognitive performance" OR "Executive function" OR "Working memory" OR "Shifiting") |
| | #4 TS = ("Randomized controlled trial" OR "Randomized" OR "Controlled" OR "Trial") |
| | #5 #1 AND #2 AND #3 AND #4 |
| CNKI | (运动 exercise + 有氧运动 aerobic exercise + 抗阻训练 resistance training + 力量训练 power training + 太极拳 Tai Chi + 瑜伽 Yoga) AND (抑郁症 Depressive disease + 抑郁 Depression) AND (认知功能 cognition function + 工作记忆 working memory + 执行功能 executive function + 刷新 update + 认知 cognition) |
| Wanfang | (运动 exercise OR 有氧运动 aerobic exercise OR 抗阻训练 resistance training OR 力量训练 power training OR 太极拳 Tai Chi OR 瑜伽 Yoga) AND (抑郁症 Depression disease OR 抑郁 Depression) AND (认知功能 cognition function OR 工作记忆 working memory OR 执行功能 executive function OR 刷新 update OR 认知 cognition) |

## Data extraction

Two researchers utilized a pre-developed spreadsheet in Microsoft Excel for information and data extraction. The heterogeneity in data extraction between the two researchers was 96.42%. The extracted information included basic information (author, year, age, sample size), experimental characteristics (exercise form, exercise intensity, exercise frequency, exercise cycle and exercise duration) and outcome indicators. Original authors were contacted by email if data were missing or unclear. When the information extracted by two researchers was inconsistent, a third researcher would join in to make a joint decision.

The coding was based on exercise type, exercise intensity, exercise duration, and intervention content. Exercise types were coded as aerobic exercise, resistance exercise, Yoga, and Taichi. Exercise intensity was coded as Low, Moderate, and Moderate-to-vigorous (mixed). Exercise duration was coded as 3–12 weeks and 13–16 weeks. Intervention content was coded as exercise only and exercise combined with other therapies (*Ren et al., 2023*).

## Quality evaluation of literature

Two researchers independently evaluated the quality of the literature using PEDro scale (Physiotherapy Evidence Database) (*Verhagen et al., 1998*), with one point for each item and a total of 10 points. If two researchers encountered disagreements, a third researcher would join in the discussion and make a joint decision.

## Quality evaluation of outcome evidence

GRADEpro software was adopted to evaluate the quality evaluation of outcome evidence. "High": very confident that the predicted value is close to the true value; "Medium": moderate confidence in the predicted value, which may be close to the genuine value, but may also be very different; "Low": limited confidence in the predicted value, which may be very different from the genuine value; "Very low": little confidence in the predicted value, which is most likely very different from the true value.

## Statistical methods

Statistical analyses were conducted using the Metafor Package in R 4.4.1 (*R Core Team, 2024*), following methods outlined in the tutorial on fitting three-level meta-analytic models of *Assink & Wibbelink (2016)* and *Cui et al. (2024)*. Given that this study includes multiple effect sizes from one individual article, which violates the assumption of independence in traditional meta-analysis methods, a random-effects three-level analysis was employed. Three-level meta-analysis allows for the decomposition of variance into three sources: sampling variance (Level 1), within-study variance (Level 2), and between-study variance (Level 3).

Pooled effect sizes using Hedges'g and 95% Confidence Intervals (CIs) of exercise on EF, as well as on its subdomains, were calculated using restricted maximum likelihood estimation. One-tailed likelihood ratio tests were used to assess statistical significance of the level 2 and level 3 variance. This study employed Hedge's g as the effect size measure, where small, medium, and large effect sizes correspond to Hedge's g values of 0.20, 0.50, and 0.80, respectively (*Cohen, 1992*).

Publication bias was assessed using a funnel plot and Egger's test. In the presence of publication bias, trim-and-fill methods were employed to adjust to any potential bias.

# RESULTS

## Results of literature retrieval

A total of 4,194 articles were obtained by searching six databases and relevant means. Firstly, 285 repeated articles were eliminated by using Endnote X9. Secondly, 3,807 articles

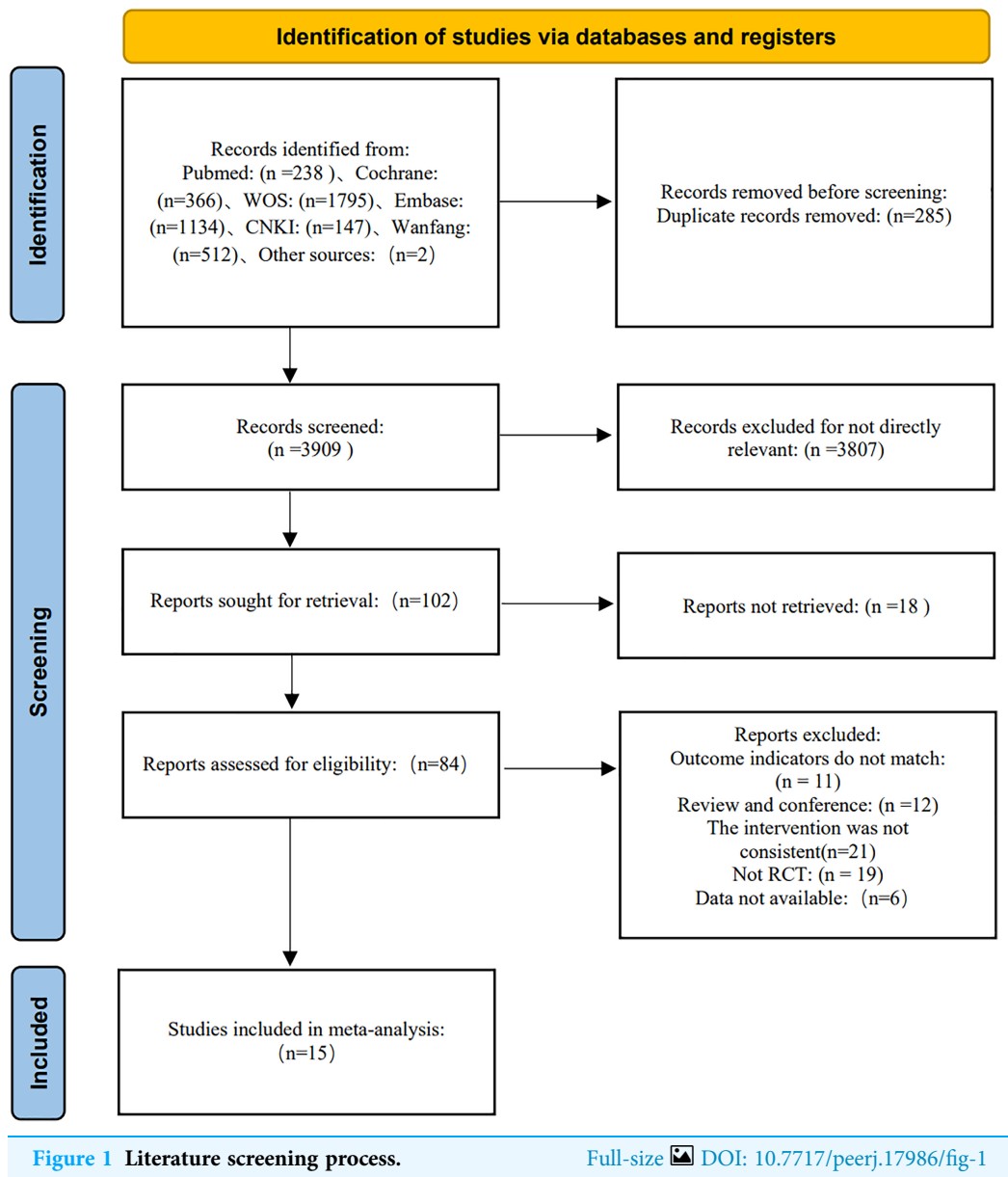

**Figure 1 Literature screening process.**

were excluded by reading their titles and abstracts. Next, of the full-text reading of 102 articles, 18 articles could not be found and 11 did not match the outcome indicators, 12 articles were literature review, 21 articles had inconsistent intervention content, 19 articles had inconsistent research design, and six articles could not extract data. Finally, 15 articles were included (*Buschert et al., 2019*; *Chen et al., 2021*; *Halappa et al., 2018*; *Hoffman et al., 2008*; *Imboden et al., 2020*; *Khatri et al., 2001*; *Krogh et al., 2014, 2009, 2012*; *Lavretsky et al., 2022*; *Oertel-Knöchel et al., 2014*; *Sharma et al., 2006*; *Zhang & Chen, 2019*; *Zhang et al., 2022*; *Zheng et al., 2019*), as shown in Fig. 1.

## Characteristics of included literature

A total of 15 articles on the intervention effect of exercise on working memory in patients with depression were included, published from 2001 to 2022, including 11 English literature and two Chinese literature, with a total of 1,067 patients, aged 30–69 year-old. Exercise forms included yoga combined with traditional therapy, aerobic exercise, resistance training, Tai Chi, aerobic exercise combined with conventional therapy and Tai Chi combined with drug therapy, *etc*. The exercise frequency was 2–5 days/week, and the exercise cycle was 3–16 weeks. The exercise intensity was divided into low intensity (*Ren et al., 2023*), moderate intensity and moderate-to-high intensity (*Garber et al., 2011*). The outcome measures were mainly DST. As shown in Table 3.

## Quality evaluation of included literature

All 15 included articles achieved "eligibility criteria", "baseline similarity", "inter-group statistical analysis" and "point measurement and variation". Two articles did not describe the method of "random allocation". One article reported "allocation concealment", three did not report "exercise load control", five did not report "outcome assessment blinding", four did not report patient withdrawal rate ≥15%, and six did not achieve "ITT (intention-to-treat analysis)". The literature scores ranged from 6 to 9, with an average score of 7.73, indicating that the methodological quality was generally good, as shown in Table 4.

## Results of meta-analyses

The results of the three-level meta-analysis model indicated a significant improvement in working memory among patients with depression due to exercise, based on 15 studies consisting of 38 effect sizes (g = 0.16, 95% CI [0.03–0.28], $p = 0.02$). This effect size is considered small. Please refer to Fig. 2 for graphical representation, as shown in Fig. 2.

The significance of within-study variance (Level 2) and between-study variance (Level 3) was assessed using one-sided likelihood ratio tests. The findings reveal no significant difference in within-study variance (Level 2) (LRT = 0.00, $p = 0.50$), while there was a significant difference in between-study variance (Level 3) (LRT = 3.09, $p = 0.04$).

Regarding the total variance components, sampling variance (Level 1) accounted for 73.51%, within-study variance (Level 2) accounts for 2.70%, and between-study variance (Level 3) accounted for 26.49%.

## Publication bias test

The funnel plot exhibits significant asymmetry on both sides. Egger's test indicated evidence of publication bias (t = 2.52, $p = 0.02 < 0.05$), suggesting a presence of publication bias in the studies. Trim-and-fill analysis was performed on the left side of the funnel plot, yielding an adjusted effect size of 0.09 with a 95% CI [−0.01 to 0.19]. Therefore, it was advised to interpret the results of this study cautiously due to the identified publication bias, as shown in Fig. 3.

**Table 3 Characteristics of included literature.** AF for animal fluency; BNT for Boston naming test; BSRT for Buschke's selective reminding test; DST for digit span test; SS for spatial span; FAS for controlled oral word association test; LNS for letter number span; TOL for Tower of London.

| Included literature | Number (E/C) | Age (E/C) | Feature of intervention | Outcome measure |
|---|---|---|---|---|
| Sharma et al. (2006) | 15/15 | 31.87 ± 8.78/ 31.67 ± 8.46 | E: Yoga combined with traditional, 3 day/week, 30 min, low, 8 week; C: Traditional therapy | DST |
| Khatri et al. (2001) | 42/42 | 56.73 ± 6.45 | E: Aerobic exercise, 3 day/week, 30 min, 70–85% HRR/moderate-to-vigorous, 16 week; C: Sertraline | DST |
| Imboden et al. (2020) | 22/20 | 38.3 ± 13.4/ 41.3 ± 9.2 | E: Aerobic exercise, 3 day/week, 45 min, 60–75% HRmax/moderate, 6 week C: Stretching | Sequence of numbers |
| Hoffman et al. (2008) | 104/49 | 51.7 ± 7.6 | E: Aerobic exercise, 3 day/week, 45 min, 70–85% HRR/moderate-to-vigorous, 16 week; C: Placebo | DST |
| Buschert et al. (2019) | 18/20 | 47.27 ± 6.84/ 47.47 ± 8.47 | E: Aerobic exercise, 2–3 day/week, 30 min, 85% HRmax/moderate-to-vigorous, 3–4 week; C: Occupational or art therapy | DST |
| Krogh et al. (2009) | 55/55 | 38.1 ± 9.0/ 36.7 ± 8.7 | E: Aerobic exercise, 2 day/week, 90 min, 70–89% HRmax/moderate-to-vigorous, 16 week; C: Relaxation | DST |
| Krogh et al. (2009) | 55/55 | 41.9 ± 8.7/ 36.7 ± 8.7 | E: Resistance resistance, 2 day/week, 90 min, 50–75% RM/moderate-to-vigorous, 16 week; C: Relaxation | DST |
| Krogh et al. (2012) | 56/59 | 39.7 ± 11.3/ 43.4 ± 11.2 | E: Aerobic exercise, 3 day/week, 45 min, 65–80% VO$_2$max/moderate-to-vigorous, 12 week; C: Stretching | DST |
| Zhang et al. (2022) | 20/19 | 47.2 ± 6.99/ 54.16 ± 6.09 | E: Taichi, 2 day/week, 90 min, low ,6 week and 12 week; C: Waiting group | TOL |
| Zheng et al. (2019) | 30/30 | 37.50 ± 9.12/ 35.17 ± 5.93 | E: Aerobic exercise combined with conventional therapy, 5 day/week, 30 min, moderate, 4 week; C: Conventional therapy | Breadth of vision |
| Chen et al. (2021) | 63/62 | 30.3 ± 7.5/ 32.7 ± 6.5 | E: Aerobic exercise combined with conventional therapy, 3 day/week, 30–60 min, 64–76% HRR/moderate, 16 week; C: Conventional therapy | DST |
| Lavretsky et al. (2022) | 89/89 | 69.2 ± 6.9/ 69.4 ± 6.2 | E: Tai chi combined with drug treatment, 3 day/week, 60 min, low, 12 week; C: Health education combined with drug treatment | FAS+AF +BNT |
| Oertel-Knöchel et al. (2014) | 8/6 | 36.63 ± 12.91/ 41.37±15.69 | E: Aerobic exercise combined with cognitive training, 3 day/week, 45 min, 60–70% HRmax/moderate, 4 week, SS+LNS; C: Cognitive training combined with relaxation training | SS+LNS |
| Krogh et al. (2014) | 41/38 | 38.9 ± 11.7/ 43.8 ± 12.2 | E: Aerobic exercise, 3 day/week, 45 min, 80% HRmax/moderate-to-vigorous, 12 week; C: Stretching | BSRT |
| Halappa et al. (2018) | 26/16/23 | 33.81 ± 10.77/ 37.06 ± 8.08/ 30.96 ± 5.94 | E1: Yoga with medication, 3 day/week, 60 min, 12 week; E2: Yoga, 3 day/week, 60 min, 12 week C: Medication | DST |
| Zhang & Chen (2019) | 22/20 | 31.4 ± 7.2/ 32.2 ± 7.6 | E: Aerobic exercise combined with conventional therapy, 3 day/week, 3 km, 4, 8 week; C: Conventional therapy | DST |

## Subgroup analyses

The commonalities in exercise frequency and the lack of exercise duration data made grouping infeasible. Therefore, subgroup analyses were conducted only in the aspect of

**Table 4 Evaluation of methodological quality of included literature.**

| Included literature | 1 | 2 | 3 | 4 | 5 | 6 | 7 | 8 | 9 | 10 | TS |
|---|---|---|---|---|---|---|---|---|---|---|---|
| *Sharma et al. (2006)* | 1 | 1 | 0 | 1 | 0 | 1 | 1 | 1 | 1 | 1 | 8 |
| *Khatri et al. (2001)* | 1 | 1 | 0 | 1 | 1 | 1 | 1 | 1 | 1 | 1 | 9 |
| *Imboden et al. (2020)* | 1 | 1 | 0 | 1 | 1 | 1 | 0 | 0 | 1 | 1 | 7 |
| *Hoffman et al. (2008)* | 1 | 1 | 0 | 1 | 1 | 1 | 1 | 1 | 1 | 1 | 9 |
| *Buschert et al. (2019)* | 1 | 1 | 0 | 1 | 1 | 0 | 0 | 0 | 1 | 1 | 6 |
| *Lavretsky et al. (2022)* | 1 | 1 | 0 | 1 | 0 | 1 | 0 | 0 | 1 | 1 | 6 |
| *Krogh et al. (2009)* | 1 | 1 | 0 | 1 | 1 | 1 | 1 | 0 | 1 | 1 | 8 |
| *Krogh et al. (2012)* | 1 | 1 | 1 | 1 | 1 | 1 | 0 | 1 | 1 | 1 | 9 |
| *Zhang et al. (2022)* | 1 | 1 | 0 | 1 | 0 | 0 | 1 | 1 | 1 | 1 | 7 |
| *Zheng et al. (2019)* | 1 | 1 | 0 | 1 | 1 | 0 | 1 | 1 | 1 | 1 | 8 |
| *Chen et al. (2021)* | 1 | 1 | 0 | 1 | 1 | 0 | 1 | 1 | 1 | 1 | 8 |
| *Oertel-Knöchel et al. (2014)* | 1 | 1 | 0 | 1 | 1 | 1 | 1 | 1 | 1 | 1 | 9 |
| *Krogh et al. (2014)* | 1 | 1 | 0 | 1 | 1 | 1 | 1 | 1 | 1 | 1 | 9 |
| *Zhang & Chen (2019)* | 1 | 0 | 0 | 1 | 1 | 0 | 1 | 0 | 1 | 1 | 6 |
| *Halappa et al. (2018)* | 1 | 0 | 0 | 1 | 1 | 1 | 1 | 0 | 1 | 1 | 7 |

exercise intensity, exercise cycle, exercise type, and intervention content, as shown in Table 5.

The type of exercise did not significantly moderate the intervention effect on working memory in patients with depression ($F(3,34) = 1.99$, $p = 0.13$). Specifically, aerobic exercise did not show statistical significance ($p = 0.33$), nor did resistance exercise ($p = 0.43$). However, Tai Chi demonstrated statistical significance ($p = 0.01$), and yoga also showed statistical significance ($p = 0.04$).

The cycle of exercise did not significantly moderate the intervention effect on working memory in patients with depression ($F(1,36) = 0.05$, $p = 0.83$). To illustrate, the "3–8 weeks" duration did not show statistical significance ($p = 0.13$), while the "12–16 weeks" duration demonstrated statistical significance ($p = 0.03$).

Exercise intensity moderated the intervention effect on working memory in patients with depression ($F(2,35) = 8.83$, $p < 0.01$). Specifically, both low intensity and moderate intensity showed statistical significance ($p < 0.01$ for both), while moderate-to-vigorous intensity did not show statistical significance ($p = 0.39$).

Intervention content did not moderate the intervention effect on working memory in patients with depression ($F(1,36) = 1.60$, $p = 0.22$). Specifically, the "Exercise only" group did not show statistical significance ($p = 0.22$), whereas the "Exercise combined with other therapies" group demonstrated statistical significance ($p = 0.01$).

## Evaluation of outcome evidence

The results were evaluated using GRADEpro and were found to be moderate, as shown in Table 6.

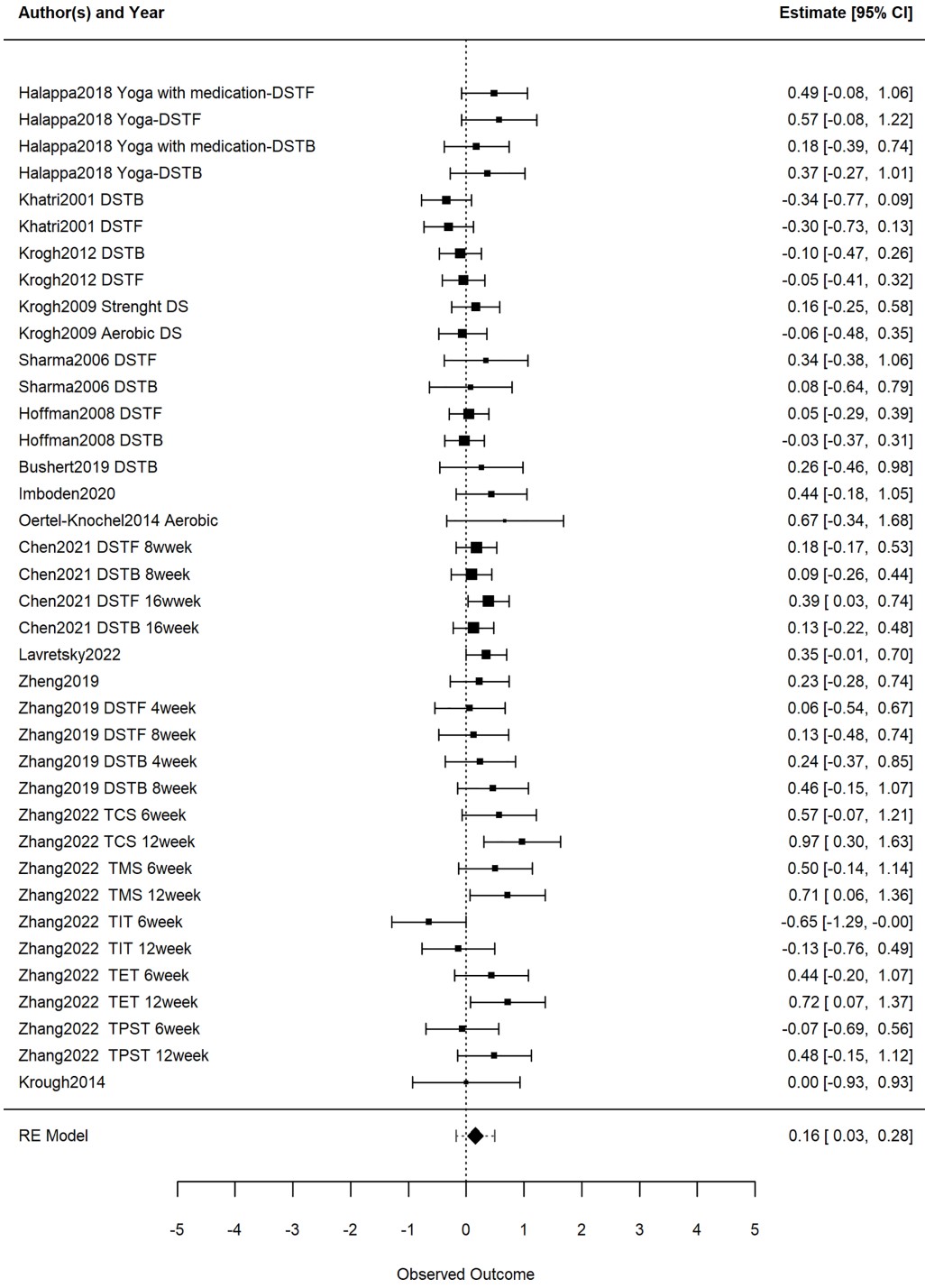

**Figure 2 Intervention exercise effects on working memory in patients with depression.** DS for digit Span; DSTB for digit Span Test Backward; DSTF for Digit Span Test Forward; TMS for Total Move Score; TCS for Total Correct Score; TIT for Total Initial Time; TET for Total Executive Time; TPST for Total Problem-Solving Time.

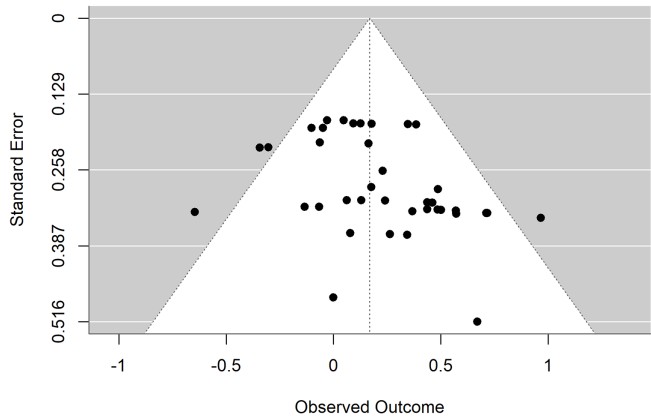

**Figure 3 Funnel plot of the intervention effect of exercise on working memory in patients with depression.**

**Table 5 Subgroup analyses of the effect of exercise intervention on working memory in patients with depression.**

| Moderating variable | Test of moderators | N(ES) | Hedge's g, 95%CI | p |
|---|---|---|---|---|
| Type of exercise | $F_{(3,34)} = 1.93$, $p = 0.13$ | | | |
| Aerobic exercise | | 20 | 0.06, [−0.07 to 0.20] | 0.33 |
| Resistance exercise | | 1 | 0.19, [−0.29 to 0.66] | 0.43 |
| Taichi | | 11 | 0.35, [0.10–0.59] | 0.01 |
| Yoga | | 6 | 0.34, [0.02–0.65] | 0.04 |
| Exercise cycle (weeks) | $F_{(1,36)} = 0.05$, $p = 0.83$ | | | |
| 3–8 | | 17 | 0.14, [−0.04 to 0.33] | 0.13 |
| 12–16 | | 21 | 0.17, [0.02–0.32] | 0.03 |
| Intensity of exercise | $F_{(2,35)} = 8.83$, $p < 0.01$ | | | |
| Low | | 17 | 0.35, [0.20–0.50] | <0.01 |
| Moderate | | 11 | 0.23, [0.08–0.37] | <0.01 |
| Moderate-to-Vigorous | | 10 | −0.06, [−0.20 to 0.08] | 0.39 |
| Intervention content | $F_{(1,36)} = 1.60$, $p = 0.22$ | | | |
| Exercise only | | 22 | 0.10, [−0.06 to 0.25] | 0.22 |
| Exercise combined with other therapies | | 16 | 0.25, [0.06–0.43] | 0.01 |

# DISCUSSION

This study included a total of 15 articles and examined 38 effect sizes to explore the effects of exercise on working memory in patients with depression. However, due to the presence of publication bias, the evidence grade was downgraded by one level and was classified as "moderate". All the included articles performed randomized controlled trials, and the average score of PEDro was 7.73. The quality of the literature was good, but only a few literature realized allocation concealment and evaluation blinding. In addition, due to the limitation of the included studies, this study did not carry out a more detailed division of the degree of depression in depressive patients, which may also affect the accuracy of the

**Table 6 Level of evidence for outcome indicators.** 1 for study limitation; 2 for inconsistency; 3 for indirectness; 4 for imprecision; 5 for publication bias; A for Downgrade 1 Level; B for Not Downgraded; E for Experimental group; C for Control group; An asterisk (*) indicates that our study was subject to publication bias.

| Outcome | RCTs | Evaluation of evidence quality level | | | | | Relative effect size | Level |
|---------|------|---|---|---|---|---|---------------------|-------|
| | | 1 | 2 | 3 | 4 | 5 | | |
| Working memory | 15 | B | B | B | B | A* | 0.16%, 95% CI [0.03–0.28] | Moderate |

results, which may be the reason for the moderate level of evidence for the results of this study. In the future, researches should better control allocation concealment, blind outcome evaluation, increase sample size and control attrition rate to improve the quality of research methodology.

The results of this study shows that exercise can improve working memory in patients with depression, with an effect size of 0.16 belonging to small effect size. Previous studies also showed similar results (*Contreras-Osorio et al., 2022*; *Ren et al., 2023*). Studies have found that exercise can improve the brain structure and function of prefrontal lobe and hippocampus, which are related to working memory. Patients with major depressive disorder have abnormal volumes of the prefrontal cortex, temporal lobe, and cingulate cortex, and depressive patients have reduced hippocampus volume. Exercise can increase the gray matter volume of frontal cortex, cingulate gyru and other cortex, reduce the apoptotic rate of hippocampal cells, promote the growth of hippocampus, increase the volume of hippocampal cortex area, improve the abnormal brain structure and function of patients with depression, so as to improve the memory ability of patients (*Baek et al., 2012*; *Brocardo et al., 2012*).

We found that exercise intensity serves as a moderating variable in the exercise intervention for improving working memory in patients with depression. Both low and moderate exercise intensities are beneficial for preserving working memory in these patients. *Jiang et al. (2022)* similarly reported that moderate-intensity aerobic exercise enhances visual learning, memory, and executive functions in individuals with depression, with improvements primarily noted in visual learning and memory as exercise intensity increases. Conversely, *Sun et al. (2018)* observed that low-intensity exercise improves cognitive function in depression, whereas moderate and high-intensity exercises do not enhance cognitive function in these individuals. According to the inverted U-shaped theory, moderate-intensity exercise yields superior improvements in cognitive function compared to low and high-intensity exercises (*Wang & Zhou, 2014*). However, *Ren et al. (2023, 2024)* reported that low-intensity exercise does not improve executive function in depression, moderate-intensity exercise does not enhance cognitive function, while moderate-to-high intensity exercise can improve both executive and cognitive functions in individuals with depression. The discrepancies may stem from the following: (1) Our study assessed only working memory, while previous studies evaluated overall executive and cognitive functions without separately dividing working memory through exercise

intensity; (2) our differing approach to categorizing exercise intensity compared to previous studies, which all classified exercise intensity into two levels, whereas our study refined it into three levels for higher precision.

We also found that exercise type, cycle, and intervention content do not act as moderating variables in the exercise intervention for improving working memory in patients with depression. Among different exercise types, Tai Chi and yoga were identified as capable of enhancing working memory, exhibiting moderate to small effect sizes. Tai Chi and yoga possess unique advantages compared to other forms of exercise, featuring relaxation, gentle movements, and mindfulness practices that integrate body, mind, and external environment, promoting dynamic balance between them and alleviating negative emotions (*Lou & Liu, 2018*; *Yang et al., 2019*). Moreover, Tai Chi and yoga involve various cognitive aspects such as attention, visual-spatial skills, and memory during practice to maintain stable body posture, which benefits cognitive functions including perception and memory improvement (*Cai et al., 2021*; *Zou et al., 2019*). Regarding exercise cycle, significant differences were observed in the 12–16 weeks category, demonstrating a small effect size. *Jiang et al. (2022)* suggested that aerobic exercise lasting 4–8 weeks improves executive functions in depression but does not enhance visual learning and memory. In contrast, aerobic exercise lasting 12–16 weeks enhances visual learning and memory but does not improve executive functions (*Jiang et al., 2022*). *Ren et al. (2023)* also found that exercise lasting more than 13 weeks improves executive functions in depression, whereas 3–12 weeks of exercise showed no difference compared to controls. The cardiovascular fitness hypothesis posits that long-term exercise enhances executive functions by improving cardiovascular health (*Voss et al., 2013*). Long-term physical exercise increases oxygen saturation and cerebral blood flow in regions such as the frontal and temporal lobes, which are directly related to cognitive functions (*Cai & Zhang, 2019*). Regarding intervention content, we found that exercise alone does not improve working memory in patients with depression, whereas exercise combined with other therapies does enhance working memory. *Sun et al. (2018)* similarly indicated that exercise alone does not improve attention, memory, or overall cognitive function in depression. Studies by *Greer et al. (2015)* and *Olson et al. (2017)* also demonstrated that combining antidepressants with moderate exercise more effectively improves cognitive symptoms in patients with depression.

## CONCLUSIONS

To sum up, exercise can improve the working memory of patients with depression; its moderating effect is the best when having low-intensity and moderate-intensity; and exercise type, exercise cycle and intervention content have no effect on the intervention effect. This study has the following limitations: (1) a total of 15 literature were included in this study, all of which were in English and Chinese, so the comprehensiveness of the included literature was limited to a certain extent; (2) among the included studies, relatively few studies had allocation concealment and blinded outcome assessment, which may affect the accuracy of the results; (3) the course of disease and degree of depression were not classified in this study, which may affect the reliability of the results, and the

results need to be treated with caution; (4) the evidence level of the results was moderate, which may have changed the results as more evidence became available.

### Funding
The authors received no funding for this work.

### Competing Interests
The authors declare that they have no competing interests.

### Author Contributions
- Cong Liu conceived and designed the experiments, performed the experiments, analyzed the data, prepared figures and/or tables, authored or reviewed drafts of the article, and approved the final draft.
- Rao Chen performed the experiments, analyzed the data, prepared figures and/or tables, and approved the final draft.
- So Mang Yun performed the experiments, analyzed the data, prepared figures and/or tables, and approved the final draft.
- Xing Wang conceived and designed the experiments, analyzed the data, authored or reviewed drafts of the article, and approved the final draft.

### Data Availability
  The data are available in the Supplemental File.

### Supplemental Information
Supplemental information for this article can be found online at http://dx.doi.org/10.7717/peerj.17986#supplemental-information.

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
