# Peer review of "Intervention effect of exercise on working memory in patients with depression: a systematic review"

_PeerJ, doi:10.7717/peerj.17986_

## Round 0.1 · original submission · Major Revisions

As you can see, the reviewers have offered constructive feedback that I believe will greatly assist you in revising your manuscript. I kindly request that you provide comprehensive responses to each comment from the reviewers.

Reviewer 1 ·

Basic reporting

In the "Introduction," final paragraph: I suggest incorporating into your analysis the prior reviews by Brondino et al., 2017, and Contreras-Osorio et al., 2022.

Brondino N, Rocchetti M, Fusar-Poli L, Codrons E, Correale L, Vandoni M, Barbui C, Politi P. A systematic review of cognitive effects of exercise in depression. Acta Psychiatr Scand. 2017 Apr;135(4):285-295. doi: 10.1111/acps.12690. Epub 2017 Jan 22. PMID: 28110494.
Contreras-Osorio F, Ramirez-Campillo R, Cerda-Vega E, Campos-Jara R, Martínez-Salazar C, Reigal RE, Hernández-Mendo A, Carneiro L, Campos-Jara C. Effects of Physical Exercise on Executive Function in Adults with Depression: A Systematic Review and Meta-Analysis. Int J Environ Res Public Health. 2022 Nov 18;19(22):15270. doi: 10.3390/ijerph192215270. PMID: 36429985; PMCID: PMC9690406.

Introduction: I suggest conducting a more comprehensive analysis regarding the distinguishing elements compared to previous systematic reviews. This will enable the reader to more clearly identify the scientific novelty of the proposal.

Experimental design

Inclusion criteria:
In this section and the following one (inclusion criteria and exclusion criteria), I suggest stating the criteria but not presenting the results, as this is part of the results section. Please justify the minimum duration of the intervention and declare the definition used for physical exercise. It is not clear whether acute, chronic interventions, or both were included. Additionally, I believe it is necessary to specify whether the intervention should solely involve exercise or could entail exercise along with another type of therapy, such as cognitive therapy (since Oertel-Knochel et al. 2014 was included).

Literature retrieval strategy: Please clarify which filters were used in the text search.

Validity of the findings

no comment.

Additional comments

I appreciate the opportunity to review your manuscript. I suggest some areas for improvement that I would appreciate you considering to enhance readers' understanding. Additionally, I believe it is crucial to highlight the scientific novelty of this systematic review in light of previous reviews. I urge the authors to provide thorough justification for this aspect.

·

Basic reporting

The manuscript is generally well-written, but some sections could benefit from a more detailed explanation, especially in describing the methodologies used in the analyzed studies. Additionally, ensure that all figures and tables are correctly labeled and described within the text. A thorough proofreading to correct minor typographical errors is also recommended.

Experimental design

The research question is relevant and well-defined. The methods are appropriate for a systematic review. However, the manuscript could enhance transparency by including more detailed information on the selection process and criteria for the included studies. This would strengthen the reproducibility of the review.

Validity of the findings

The conclusions are appropriately cautious given the moderate evidence level reported. However, it would be beneficial to discuss the implications of these findings in broader contexts, such as potential impacts on clinical practices or guidelines. Consider also discussing the limitations in more depth, particularly those related to the heterogeneity of exercise interventions and outcome measures.

Additional comments

This review is timely and contributes to the understanding of non-pharmacological interventions for depression. Future research directions could include more diverse populations and standardized intervention protocols to enhance the generalizability of the results. Also, consider discussing the potential mechanisms through which exercise impacts cognitive functions in depressed individuals, linking it to existing literature.

Reviewer 3 ·

Basic reporting

1) The introduction does not provide information on bout the relevance of the present work and its incremental contribution to previous meta-analyses. Although the authors refer to two prior meta-analyses (Rent et al., 2023; Sun et al., 2018) that examined the effect of exercise on working memory, there is no discussion of their conflicting results. Another meta-analysis by Contreras-Osorio et al. (2022) is not discussed at all. Importantly, the present meta-analysis is a simple replication of Ren et al. (2023). Exactly the same studies considered in Ren et al. (2023) are included in the present analysis. It remained unclear what the reanalysis adds to what is already known.

2) The text is sometimes confusing and unclear. For example, what is meant by the statement that “it is supplemented by tracing the relevant systematic reviews and references of the included literature” (page 8, line 90)? What is meant by the statement that depression is a “disability”? In some places whole sentences are not even used (e.g., page 7, line 80). Furthermore, the authors make some strong claims without supporting them. For example, what is the evidence that depression has high a mortality rate (page 5, line 21)? The cited reference (Malhi & Mann, 2018) does not provide this information.

3) Although the authors provided some data that was presumably used for the meta-analysis as supplemental file, no codebook was provided. Therefore, the meaning of the unlabelled data columns is unclear.

Experimental design

4) The description of the meta-analytic approach needs to be improved. For example, what estimator was used for the random-effects model to estimate between-study heterogeneity? How were effect sizes calculated (e.g., Cohen’s d, Hedge’s g)? How was the moderator of exercise frequency grouped into low versus moderate/high intensity (i.e., what were the cut-offs)? How was publication bias assessed? The methods section should be substantially revised to report the methodological details of the meta-analysis. Otherwise, it is not possible to replicate the analyses.

5) The authors seem to have coded several moderating variables such as exercise type, exercise time, exercise period, and exercise intensity (page 10, line 158). However, no information was provided describing the meaning of these variables and how they were measured.

6) Although two raters independently extracted all information from the primary studies, no information on the coding quality was provided. Therefore, I recommend reporting relevant intercoder statistics for the coded variables and study quality ratings.

7) The authors should report the calculated effect sizes and standard errors for each study.

Validity of the findings

8) For primary studies that reported multiple effects, the authors selected only one effect size and ignored the remaining effects (page 7, line 70). This approach is not recommended because it ignores relevant information that should be part of the meta-analysis. Dependent effect sizes resulting from effects derived from the same sample can be pooled in multivariate meta-analyses, three-level meta-analytic models, or meta-analyses with robust variance estimation. The authors should consider one of the these more appropriate meta-analytic approaches and pool the entire corpus of available effects.

9) The literature search was limited to academic databases (see page 8). However, gray literature was not addressed. I do not know if there is a substantial amount of unpublished research on the investigated topic, but I recommend searching for gray literature such as conference proceedings, theses, or unpublished manuscripts to locate any not formally published research. An additional search in Google Scholar may be helpful in this respect. Limiting systematic reviews and meta-analyses to formally published research carries the risk of overestimating effects because publication bias tends to favor the publication of significant results with large effects. Therefore, reviews should usually aim to summarize the entire body of available studies on the studied topic.

10) The I2 measure is an indicator of *relative* heterogeneity, not absolute heterogeneity. Therefore, it cannot be used to decide on the meta-analytic model (i.e., random- or fixed-effects). I strongly recommend estimating and reporting the between-study heterogeneity for the meta-analyses.

11) Although only 13 studies were identified, the meta-analyses pooled 16 effect sizes. Does this mean that some studies reported multiple effect sizes? If so, wouldn’t it be necessary to acknowledge the dependencies between effect sizes?

---

## Round 0.2 · accepted · Accept

Thank you for addressing the reviewers' concerns.

Reviewer 1 ·

Basic reporting

I have reviewed all the changes made by the authors in response to the revision of their manuscript. I believe that the manuscript now meets the necessary conditions for publication.

Experimental design

no comment

Validity of the findings

no comment

Additional comments

I have reviewed all the changes made by the authors in response to the revision of their manuscript. I believe that the manuscript now meets the necessary conditions for publication.